# Effects of Ambient Temperature on Nanosecond Laser Micro-Drilling of Polydimethylsiloxane (PDMS)

**DOI:** 10.3390/mi14010090

**Published:** 2022-12-29

**Authors:** Ya Lu, Chaoran Lin, Minghui Guo, Youmin Rong, Yu Huang, Congyi Wu

**Affiliations:** 1State Key Lab of Digital Manufacturing Equipment and Technology, Huazhong University of Science and Technology, Wuhan 430074, China; 2School of Mechanical Science and Engineering, Huazhong University of Science and Technology, Wuhan 430074, China; 3Guangdong Provincial Key Laboratory of Digital Manufacturing Equipment, Guangdong HUST Industrial Technology Research Institute, Dongguan 523808, China

**Keywords:** laser micro-drilling, polydimethylsiloxane, ambient temperature, micro-hole taper, wrinkling, thermal stress

## Abstract

In this research, effects of ambient temperature (−100 °C–200 °C) on nanosecond laser micro-drilling of polydimethylsiloxane (PDMS) was investigated by simulation and experiment. A thermo-mechanical coupled model was established, and it was indicated that the top and bottom diameter of the micro-hole decreased with the decrease of the ambient temperature, and the micro-hole taper increased with the decrease of the ambient temperature. The simulation results showed a good agreement with the experiment results in micro-hole geometry; the maximum prediction errors of the top micro-hole diameter, the bottom micro-hole diameter and micro-hole taper were 2.785%, 6.306% and 9.688%, respectively. The diameter of the heat-affected zone decreased with the decrease of the ambient temperature. The circumferential wrinkles were controlled by radial compressive stress. As the ambient temperature increased from 25 °C to 200 °C, the radial compressive stress gradually decreased, which led to the circumferential wrinkles gradually evolving in the radial direction. This work provides a new idea and method based on ambient temperature control for nanosecond laser processing of PDMS, which provides exciting possibilities for a wider range of engineering applications of PDMS.

## 1. Introduction

Polydimethylsiloxane (PDMS) film is a kind of high polymer, which is widely used in microfluidics [1], flexible electronics [2], biomedicine [3] and other fields [4] due to its excellent chemical stability, permeability, biological compatibility and mechanical flexibility, as well as its low cost and convenience in preparation [5,6,7,8]. PDMS is difficult to process by traditional mechanical processing methods such as stamping. It is achievable through the lithography process, but this process is complicated and costly [9]. Compared with these processing methods, laser processing is a more efficient and convenient processing method [10], and it has been widely used for the preparation of superhydrophobic microstructures [11], bionic microstructures [12], phantom microstructures [13], etc.

PDMS after laser microstructure processing can have certain surface characteristics, such as superhydrophilicity, superhydrophobicity [14], underwater superoleophilicity, underwater superoleophobicity [15], underwater superaerophobicity [16] and so on. For example, a rough through-micro-hole-array PDMS sheet prepared by laser manufacturing can be used for selective passage of air bubbles and further collection of underwater gas [17]. The laser ablated PDMS surfaces can achieve six different types of super-wettabilities [18]. Infrared laser irradiation provides a fast yet controllable way to create wrinkled micropatterns at a low cost, which can facilitate a broad array of studies in surface engineering, cellular biomechanics and optics [19]. In addition, the microstructure can also change the overall mechanical characteristics of PDMS. In the field of flexible electronics, it is significant to improve the tensile properties of flexible electrons effectively, and the key is to improve the tensile performance of flexible substrates. PDMS is one of the most commonly used flexible substrate materials [20]; it is of great significance to study how to improve the tensile performance of PDMS. A reasonable micro-hole-arrayed structure can effectively improve the tensile performance of PDMS. By adjusting the technological parameters of laser machining, the PDMS micro-hole-arrayed structure can be better quantitatively prepared, which makes up for the disadvantage that the micro-hole-arrayed structure cannot be quantitatively prepared with natural biological materials such as rose petals [21] and lotus leaves [22].

Researchers have studied the effects of laser processing parameters such as laser wavelength, pulse duration, pulse frequency, laser fluence, and scanning speed on the processing of PDMS microstructure [23,24,25,26], and many researchers have also studied the effects of liquid-assisted and gas-assisted laser processing [27,28]. Due to the low thermal decomposition temperature of PDMS, ambient temperature is one of the critical heat sources in PDMS laser micro-hole drilling, but the effects of ambient temperature in laser processing have invariably been neglected, and no previous scholars have performed research in this regard.

In this paper, the effects of ambient temperature (−100 °C–200 °C) on the geometric characteristics and the heat affected zone (HAZ) of PDMS that has been micro-hole processed by nanosecond ultraviolet (UV, 355 nm) laser were studied. The experimental results show that under the same process parameters, lowering the ambient temperature can effectively reduce the size of HAZ and thus improve the process quality. A thermo-mechanical coupled model was established to explain the effects mechanism. The effects of ambient temperature on the geometric characteristics of the micro-hole were analyzed by solving the transient temperature field distribution and then verified through experiments. Aiming at the phenomenon that the HAZ observed in the experiment changed with the change in the ambient temperature, the evolution of the wrinkles in the HAZ is explained by solving the transient stress distribution.

## 2. Experiment

### 2.1. Preparation of PDMS

The PDMS prepolymer and curing agent purchased from Dow Corning were mixed at a weight ratio of 10:1. Then, the bubbles of the mixture were removed by vacuum negative pressure in a vacuum desiccator. The mixture was then poured onto a surface of 5-inch silicon wafer and homogenized by a spin coater at 2000 rpm for a minute. After being cured at 80 °C for 2 h and peeled off from the wafer, the PDMS film with the thickness of 100 μm was prepared. Before the nanosecond treatment, the sample was cleaned in an ultrasonic cleaning machine with ethanol for 15 min. The same cleaning process was employed after laser irradiation.

### 2.2. Laser Micro-Drilling System

Figure 1 shows the laser micro-drilling system in this study, which is composed of computer, nanosecond ultraviolet laser (Poplar-355-15A5, Huaray, WuHan, China), mirror, galvanometric scanning system (intelliSCAN III 14, SCANLAB, Puchheim, Germany), focusing lens (4401-511-000-21, LINOS, Waltham, MA, USA), temperature control platform and z-platform. The PDMS sample was placed in the temperature control platform. The galvanometric scanning system was used to control the machining position in the x–y plane, the z-platform was used to control the height of the machining position, and the focusing lens with a focal length of 167 mm was used to focus the beam onto the PDMS surface. The diameter of the focusing spot was 80 μm.

The amount of material removed with laser micro-drilling mainly depends on laser fluence and pulse duration. In this study, as shown in Table 1, the laser fluence was set as 18.87 J/cm², the pulse duration was set as 5 × 10^−4^ s, and the pulse frequency was set as 10 kHz. Different ambient temperatures were set for each group of experiments by PC and temperature controller to study the effects of ambient temperature on PDMS micro-hole processing. For the repeatability of the experiment, an experiment of the same ambient temperature was repeated 10 times in each group, and 10 PDMS samples were obtained for each group.

### 2.3. Quality Evaluation Indexes of the Micro-Hole

In this study, the evaluation was carried out around the geometric characteristics and the HAZ of the micro-hole. The top and bottom diameter of the micro-hole, micro-hole taper, and the size and morphology of HAZ were the evaluation indexes. Different micro-hole geometric characteristics and HAZ characteristics will change the material properties of PDMS, which could make it suitable for more different application scenarios. Therefore, the research focus of this work is the effects of ambient temperature on the above indexes, so as to achieve the purpose of adjusting above indexes from the perspective of ambient temperature.

As shown in Figure 2b, when determining each index value of the micro-hole, 6 cross sections with spacings of 30° of the micro-holes were randomly selected, and the top and bottom diameter of the 6 cross sections were obtained, respectively. Finally, the average value of the top and bottom diameters of the 6 cross sections was assigned to the micro-hole. For one of the cross sections shown in Figure 2, a rounded corner was formed at the top micro-hole after processing. In this study, the top micro-hole diameter was determined by Equation (1) for higher accuracy definition, and the bottom micro-hole diameter was obtained by direct measurement. After the top and bottom diameters of the 6 cross sections were calculated, respectively, the top and bottom diameter and the micro-hole taper degree can be calculated using Equations (2) and (3), respectively. As shown in Figure 2c, the diameters of 6 directions with an interval of 30° randomly selected in HAZ were summed and averaged, and the difference between the average value and the top micro-hole diameter ($d_{t}$) was taken as the diameter of HAZ. Equation (4) is the calculation method. For each ambient temperature, 10 micro-holes were selected for each index measurement and calculation, and the final values of each index at each ambient temperature were obtained by taking the average of these 10 micro-holes.
(1)$$d_{tn}=\left[{\left(d_{tn}\right)}_{in}+{\left(d_{tn}\right)}_{out}\right]/2$$
(2)$$d_{t}=\frac{1}{6}\sum_{n=1}^{6}d_{tn},\;d_{b}=\frac{1}{6}\sum_{n=1}^{6}d_{bn}$$
(3)$$\theta =arctg\left[\left(d_{t}-d_{b}\right)/2t\right]$$
(4)$$d_{HAZ}=\frac{1}{6}\sum_{n=1}^{6}{\left(d_{HAZ}\right)}_{n}-d_{t}$$

### 2.4. Characterization

The micro-hole morphology in the samples were observed by field emission scanning electron microscopy (SEM, Sirion 200, FEI, Hillsboro, OR, USA), laser confocal microscope (LSCM, VK-X200K408, KEYENCE, Ōsaka, Japan) and metallographic microscope (OMT-5RT, OMT, Suzhou, China). Among them, SEM was used to observe the surface details of the PDMS micro-hole. As PDMS is not conductive, a 15 nm thick gold layer was plated on the surface of the PDMS before SEM imaging. LSCM was used to observe the three-dimensional (3D) morphology of the PDMS micro-hole and to measure the top and bottom diameter of the micro-hole. The metallographic microscope was used to measure the diameter of HAZ.

## 3. Thermo-Mechanical Modeling

PDMS laser processing is a complex process involving photothermal and photochemical reactions [29]. Since PDMS is a thermosetting material, the material in the area where the thermal decomposition temperature is reached will be converted to carbides and some short-chain silicon-containing units [30]. In order to simplify the analysis process and focus more on the effects of ambient temperature on the processing results, the complicated physical process can be rationally simplified. Some assumptions are listed as follows:The laser energy distribution is assumed as a typical Gaussian distribution.When the temperature reaches the thermal decomposition temperature, the material is considered removed.Recoil pressure and the absorption of laser by the vapor and plasma are ignored.The solid-state phase transformation is ignored.The elastic model is used to calculate the thermal stress, ignoring the plastic effects.

### 3.1. Thermal Model

In order to make the simulation results as consistent as possible with the experimental results, the parameters set by the simulation should be consistent with the actual values. In this study, a fixed Gaussian laser beam was used to process PDMS. The fixed Gaussian laser fluence can be defined as follows.
(5)$$Q\left(x,y,t\right)=\frac{2Q_{0}}{\pi r^{2}}\exp\left(-2x^{2}/r_{s}{}^{2}\right)pulse\left(t_{s},\;1/f\right)$$
where *Q*_0_ is the laser fluence of the output, *r* is the spot radius, *α* is the absorption coefficient of PDMS, *t_s_* is the pulse duration, and *f* is the pulse frequency. The temperature distribution in PDMS during processing is defined as T(x,y,t). In order to obtain the temperature distribution of PDMS, the following governing equation of heat transfer model is established.
(6)$$\rho C_{p}\frac{\partial T}{\partial t}=Q\left(x,y,t\right)+k\left(\frac{\partial^{2}T}{\partial x^{2}}+\frac{\partial^{2}T}{\partial y^{2}}\right)$$
where *ρ* is the density of PDMS, and *Cp* is the specific heat of PDMS. After establishing the governing equation of heat transfer model, the equation of the initial conditions and the additional certain boundary conditions should be defined in order to obtain the only solution. Equation (7) is the initial condition equation established. Equation (8) is the boundary condition equation of the upper surface of PDMS, and the remaining surface is set as the insulation surface.
(7)$$T\left(x,y,0\right)=T_{set}$$
(8)$$-k\frac{\partial T}{\partial n}|\begin{array}{c} {}_{\Gamma } \end{array}=Q\left(x,0,t\right)-h\left(T-T_{set}\right)-\varepsilon \sigma \left(T^{4}-T_{set}{}^{4}\right)$$
where *T_set_* is the initial temperature of PDMS, that is, the ambient temperature, and its specific values are shown in Table 2; *h* is the convection coefficient; *k* is the thermal conductivity; and *ε* is the surface emissivity. In the process of laser action, the upper surface not only received the heat from the laser but also conducted convective heat transfer and heat radiation exchange with the outside world.

### 3.2. Thermal Stress Model

The heat transfer module was coupled to the solid mechanics module in the software. The mechanical analysis field is set as linear elastic material, and the stress distribution is obtained by transient temperature distribution. The relationship between stress and strain is given by the following equations [31].
(9)$$\varepsilon_{x}=\left(\sigma_{x}-v\sigma_{y}\right)/E+\alpha T$$
(10)$$\varepsilon_{y}=\left(\sigma_{y}-v\sigma_{x}\right)/E+\alpha T$$
(11)$$\gamma_{xy}=2\tau_{xy}(1+\nu )/E$$
where *ε_x_*, *ε_y_* represent normal strains of two degrees of freedom; *σ_x_*, *σ_y_* are normal stresses of two degrees of freedom, respectively; *γ_xy_* is the shear strain; *τ_xy_* is the shear stress; *v* is the Poisson’s ratio; *E* is the Young’s modulus; and *α* is the linear expansion coefficient.

### 3.3. Computation Implementation

A two-dimensional model was developed using COMSOL Multiphysics software combined with heat transfer and solid mechanics. Since the spot size of the nanosecond laser was very small, a mesh refinement was required around the laser processing position. Auto triangle elements were utilized, where the minimum mesh size was 2 μm, and the computational domain contained 17214 elements. The thermal decomposition temperature of the material was set as 346 °C. In addition, the time step of this model was set to 5 × 10^−4^ s. The material properties involved in the mechanical analysis were shown in Table 2.

## 4. Results and Discussion

### 4.1. Analysis of Temperature Distribution

The entire laser drilling process was completed within 5 × 10^−4^ s, during which time the laser energy applied to PDMS caused the temperature of PDMS to rise, and the heat gradually diffused toward the depth and the radius. Taking the ambient temperature of 75 °C as an example, Figure 3 shows the temperature field distribution of the micro-hole middle section with times of 5 × 10^−5^ s, 5 × 10^−4^ s, 0.01 s and 10 s at this ambient temperature. During the laser pulse action, the maximum temperature was located at the micro-hole wall, and the temperature gradually decreased during the transition from the micro-hole wall to the outside of the micro-hole, and the temperature was equal to the ambient temperature at the place far enough away from the micro-hole. When t = 5 × 10^−5^ s, the area that reached the thermal decomposition temperature began to decompose. When t = 5 × 10^−4^ s, the maximum temperature was 346 °C while the minimum was 75 °C. After that, the laser stopped drilling, and the PDMS was cooled at ambient temperature. When t = 0.01 s, the maximum temperature decreased rapidly to 93.7 °C, and the maximum was 75.1 °C. When t = 10 s, the maximum temperature decreased to 77.4 °C, and the minimum temperature increased to 77 °C instead. It can be observed that the heat in the high-temperature region diffused to the low-temperature region during cooling, resulting in the temperature in the PDMS gradually becoming consistent and approaching the ambient temperature.

### 4.2. The Effects of Ambient Temperature on Micro-Hole Geometry

Although each experiment had the same laser drilling parameters, the ambient temperature of each experiment was different, which meant that the PDMS in each experiment had different initial temperatures, and the energy propagation of the Gaussian pulse beam in PDMS and the heat exchange between PDMS and the outside world were not the same during the drilling process. Therefore, a micro-hole with different morphology was formed in each experiment in the same drilling time. Although the temperature was also diffused in PDMS after 5 × 10^−4^ s, the maximum temperature was no longer increased, so it could be considered that the thermal decomposition of PDMS would not occur thereafter, and the morphology of the micro-hole at 5 × 10^−4^ s would be the final morphology.

It can be intuitively seen from Figure 4 that the micro-hole geometrical characteristics of the middle section at different ambient temperatures at t = 5 × 10^−4^ s in the simulation that it was obvious that the taper of the micro-hole increased with the increase of the ambient temperature. The SEM images in Figure 5 show the surface morphology characteristics of the micro-holes at different ambient temperatures, and the LSCM images show the 3D morphology and contour curve of the micro-holes at different ambient temperatures, which can verify the simulation results shown in Figure 4.

As can be seen from Figure 6a, when the ambient temperature was −100 °C, PDMS was not drilled through. When the ambient temperature increased from −25 °C to 200 °C, both the top and bottom diameter of the micro-hole increased, but the taper of the micro-hole decreased. As can be seen from Figure 6b, the larger the ambient temperature, the smaller the decline rate of the curve in the temperature drop area, the slower the temperature transition from the ablation temperature to the ambient temperature, and the smaller the temperature gradient. When the ambient temperature was 200 °C, the transition distance from the ablative temperature to the ambient temperature was the largest, and the radius of the rounded corner at the orifice was the largest, which was consistent with the result shown in Figure 6a.

As shown in Figure 7, in the simulation, when the ambient temperature increased from −25 °C to 200 °C, the top micro-hole diameter increased from 73.93 μm to 87.52 μm, with a growth rate of 18.38%. The bottom micro-hole diameter increased from 37.22 μm to 67.94 μm, with a growth rate of 82.54%. Since the growth rate of the top micro-hole diameter was greater than that of the bottom micro-hole diameter, the micro-hole taper decreased from 0.182 to 0.097, and the decrease rate was 46.70%.

In the experiment, when the ambient temperature increased from −25 °C to 200 °C, the top micro-hole diameter increased from 76.05 μm to 85.92 μm, with a growth rate of 12.98%. The bottom micro-hole diameter increased from 36.02 μm to 67.21 μm, with a growth rate of 86.59%. Since the growth rate of the top micro-hole diameter was greater than that of the bottom micro-hole diameter, the micro-hole taper decreased from 0.182 to 0.097, and the decrease rate was 52.79%.

The maximum prediction errors (*E*_max_) of the top micro-hole diameter, bottom micro-hole diameter and micro-hole taper were 2.785%, 6.306% and 9.688%, compared with the average value of 10 sets of data for each sample, which indicated that the simulation results showed a good agreement with the experiment results in micro-hole geometry.

### 4.3. The Effects of Ambient Temperature on HAZ

#### 4.3.1. Composition and Diameter of HAZ

In order to observe the change of the HAZ with the ambient temperature, Figure 8 shows the micro-hole surface morphology in a larger field of view. It can be seen from Figure 8g that the clastic zone and wrinkle zone were wrapped around the micro-hole, which together constituted the HAZ. The clastic zone was adjacent to the micro-hole and had a small area, which was distributed with some debris and particles formed during processing. The area of the wrinkle zone is large, and the wrinkles inside could be divided into radial and circumferential wrinkles. As shown in Figure 8a–f,h, the diameter of the HAZ gradually decreased from 253.49 μm to 0 as the ambient temperature decreased from 200 °C to −100 °C, and the average diameter change rate in the process of the ambient temperature decreased from 200 °C to 25 °C (0.229 μm/°C) and was less than that of the ambient temperature decreased from 25 °C to −25 °C (1.39 μm/°C). When the ambient temperature was −100 °C, although there was basically no HAZ, PDMS was not drilled through. When the temperature was −25 °C, PDMS was drilled through, and the diameter of the HAZ was small. From this, it can be inferred that there is probably an ambient temperature between −100 °C and −25 °C in which PDMS can be drilled through and the HAZ is smallest.

#### 4.3.2. Evolution of Wrinkle Morphology

Compared with the diameter of the HAZ, we may be more concerned about the change of wrinkle morphology in the HAZ, on account of the fact that different wrinkle morphologies may bring different surface properties. In the process of laser interaction with PDMS, the uneven distribution of temperature in the PDMS promoted the generation of thermal stress. When the thermal stress reached a certain value, the PDMS would release the thermal stress by wrinkling or even breaking [32,33]. When the ambient temperature was low (−100 °C–25 °C), the slow molecular motion in the PDMS resulted in the elasticity of PDMS decreasing and the hardness increasing, which led to few wrinkles in the HAZ. This can be confirmed by Figure 8a,b. As can be seen from Figure 8a, when the ambient temperature was −100 °C, almost no wrinkles were formed in the HAZ, and only a small number of solid particles were distributed. As shown in Figure 8b, when the ambient temperature was −25 °C, the solid particles in the HAZ were increased, and a small number of wrinkles were formed. Therefore, we focus on the evolution of HAZ at higher ambient temperatures (25 °C–200 °C).

As shown in Figure 8c, when the ambient temperature was 25 °C, a large number of circumferential wrinkles surrounding the micro-hole and radial wrinkles diverging along the orifice radius were formed in the HAZ. The schematic diagram of the wrinkles at this time is shown in Figure 8i. It can be seen from Figure 8d,e that when the environmental temperature increased from 25 °C to 200 °C, the circumferential wrinkles gradually diffused to the direction of radius, forming a tree-trunk-like wrinkle. The schematic diagram of wrinkle evolution in this process is shown in Figure 8j,k. As shown in Figure 8f, when the ambient temperature reached 200 °C, the circumferential wrinkles basically disappeared. At this time, the wrinkles were basically all radial wrinkles, and the schematic diagram of the wrinkles is shown in Figure 8l.

As shown in Figure 9a–d, in the experiments at ambient temperatures of 25 °C, 75 °C, 125 °C and 200 °C, the compressive stress in the radius direction reached the maximum value when t = 5 × 10^−4^ s. Therefore, the compressive stress in the radius direction when t = 5 × 10^−4^ s should be mainly observed. As shown in Figure 10, it can be observed that at the same radius, the compressive stress in the radial direction decreased with the increase of the ambient temperature when t = 5 × 10^−4^ s. As a result, the circumferential wrinkles gradually evolved to the radial direction as the ambient temperature increased. When the ambient temperature was 200 °C, the compressive stress in the radial direction at different radii was low, and the excessively low compressive stress in the radial direction was insufficient to promote the generation of circumferential wrinkles, so there were basically no circumferential wrinkles at this ambient temperature.

## 5. Conclusions

In this paper, the effects of ambient temperature (−100 °C–200 °C) on the nanosecond UV laser micro-drilling of PDMS is studied by simulation and experiment. The following three conclusions can be drawn:

(1) The geometric characteristics of micro-holes were analyzed. The top and bottom micro-hole diameter decreased with the decrease of the ambient temperature, and the micro-hole taper increased with the decrease of the ambient temperature. The simulation results showed a good agreement with the experimental results in micro-hole geometry; the maximum prediction errors of the top micro-hole diameter, the bottom micro-hole diameter and micro-hole taper were 2.785%, 6.306% and 9.688%, respectively.

(2) The HAZ consisted of clastic zone and wrinkle zone. The diameter of HAZ decreased with the decrease of the ambient temperature, and the decrease rate (0.229 μm/°C) in the low-temperature stage (−100 °C–25 °C) was significantly greater than that (0.229 μm/°C) in the high-temperature stage (25 °C–200 °C). When the ambient temperature was reduced to a certain degree, the diameter of the HAZ was close to 0.

(3) The transient stress distribution was analyzed by the thermo-mechanical coupled model. The circumferential wrinkles were controlled by radial compressive stress. As the ambient temperature increased from 25 °C to 200 °C, the radial compressive stress gradually decreased, which led to the circumferential wrinkles gradually evolving to the radial direction.

This study provides a new idea and method based on ambient temperature control for nanosecond laser processing of PDMS or polymer materials with similar physical and chemical properties to PDMS, which has practical significance for nanosecond laser polymer processing.

## Figures and Tables

**Figure 1 micromachines-14-00090-f001:**
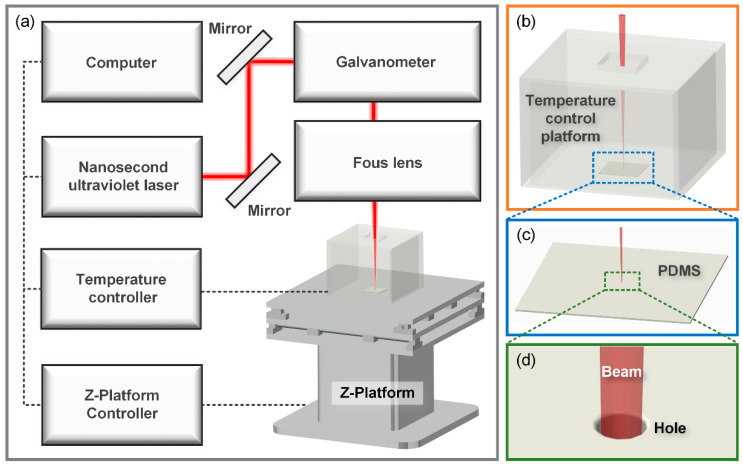
Schematic illustration of PDMS nanosecond UV laser micro-drilling system with control of ambient temperature. (**a**) Schematic diagram of laser processing control system. (**b**) Ambient temperature control platform. (**c**) PDMS samples in processing. (**d**) Enlarged view of laser micro-drilling.

**Figure 2 micromachines-14-00090-f002:**
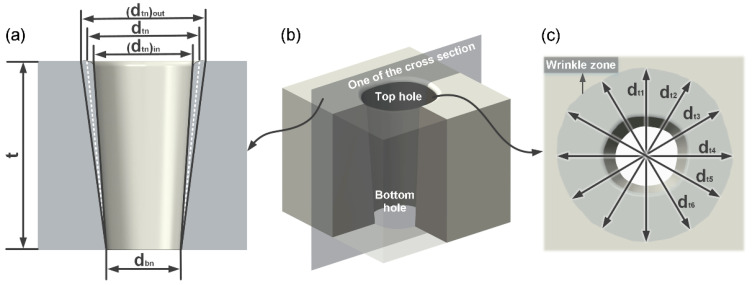
(**a**–**c**) Method of measuring the top and bottom diameter, taper and HAZ diameter of the micro-hole.

**Figure 3 micromachines-14-00090-f003:**
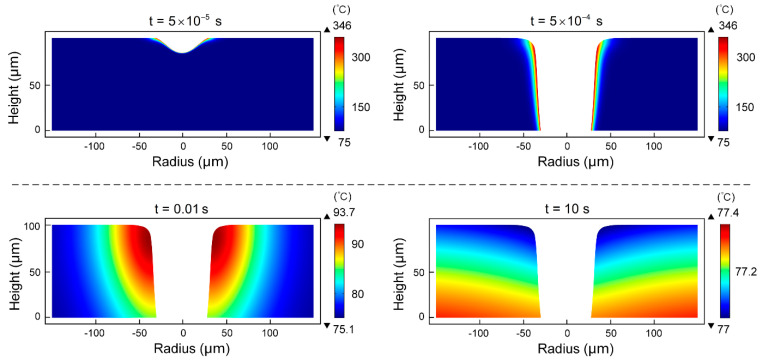
When the ambient temperature is 75 °C, the temperature field distribution of the micro-hole middle section at 5 × 10^−5^ s, 5 × 10^−4^ s, 0.01 s and 10 s, respectively.

**Figure 4 micromachines-14-00090-f004:**
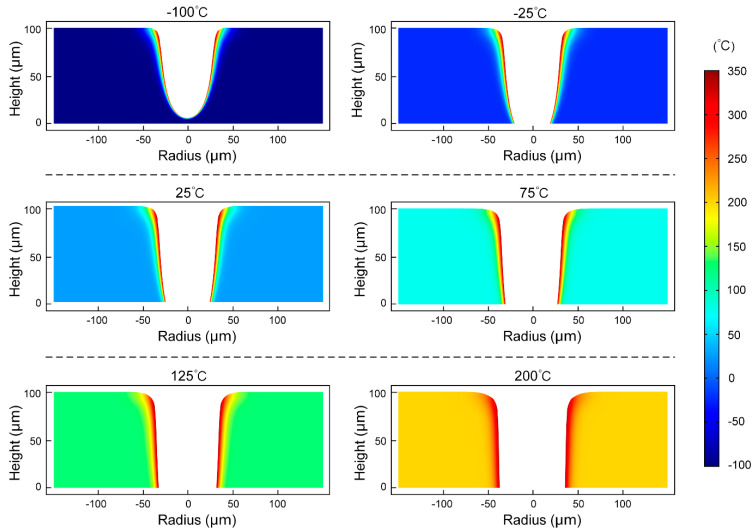
Temperature distribution of the middle section at t = 0.005 s when the ambient temperature is −100 °C, −25 °C, 25 °C, 75 °C, 125 °C and 200 °C, respectively.

**Figure 5 micromachines-14-00090-f005:**
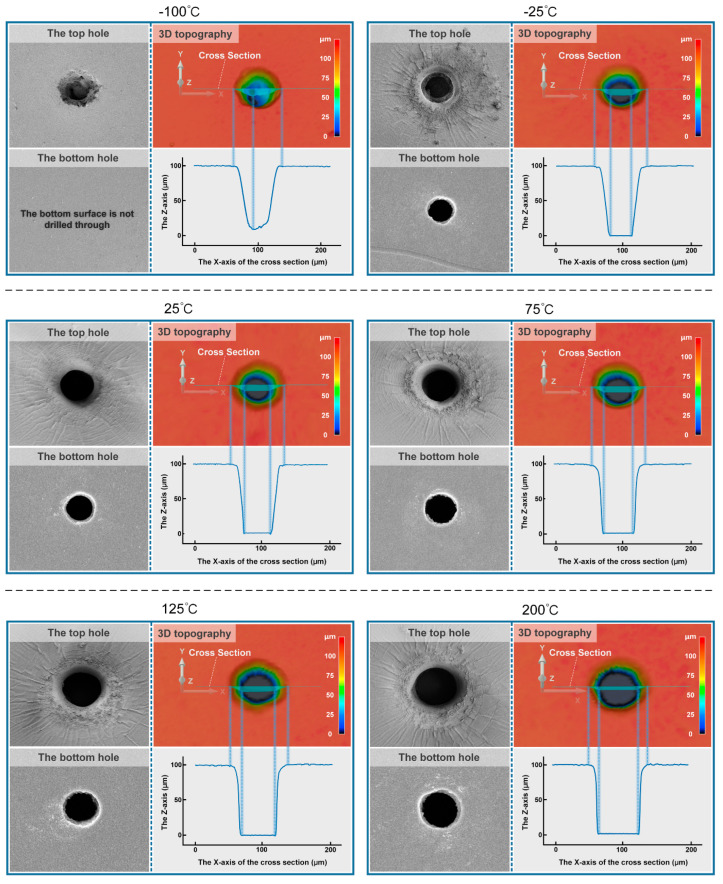
SEM images of the top and bottom micro-hole of PDMS after laser processing at different ambient temperatures. LSCM images of the 3D morphology and contour curve of the micro-hole after laser processing at different ambient temperatures.

**Figure 6 micromachines-14-00090-f006:**
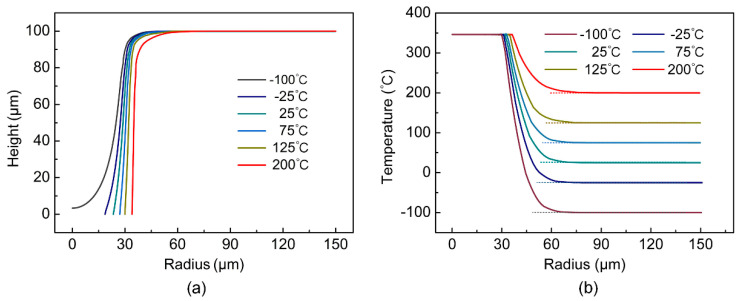
(**a**) When t = 0.0005 s, the profile of the upper surface along the radius direction at different ambient temperatures. (**b**) When t = 0.0005 s, the temperature along the radius direction of the upper surface at different ambient temperatures.

**Figure 7 micromachines-14-00090-f007:**
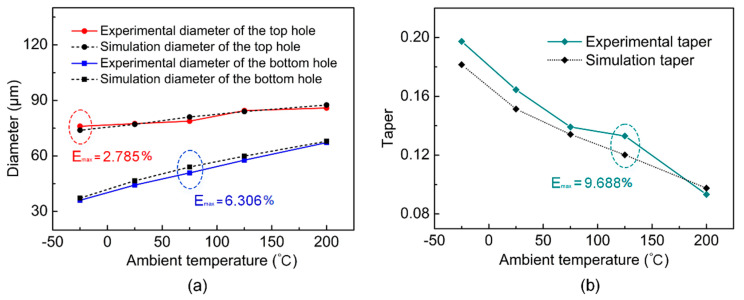
(**a**) Experimental and simulation diameter of the top and bottom micro-hole at different ambient temperatures. (**b**) Experimental and simulation taper of the micro-hole at different ambient temperatures.

**Figure 8 micromachines-14-00090-f008:**
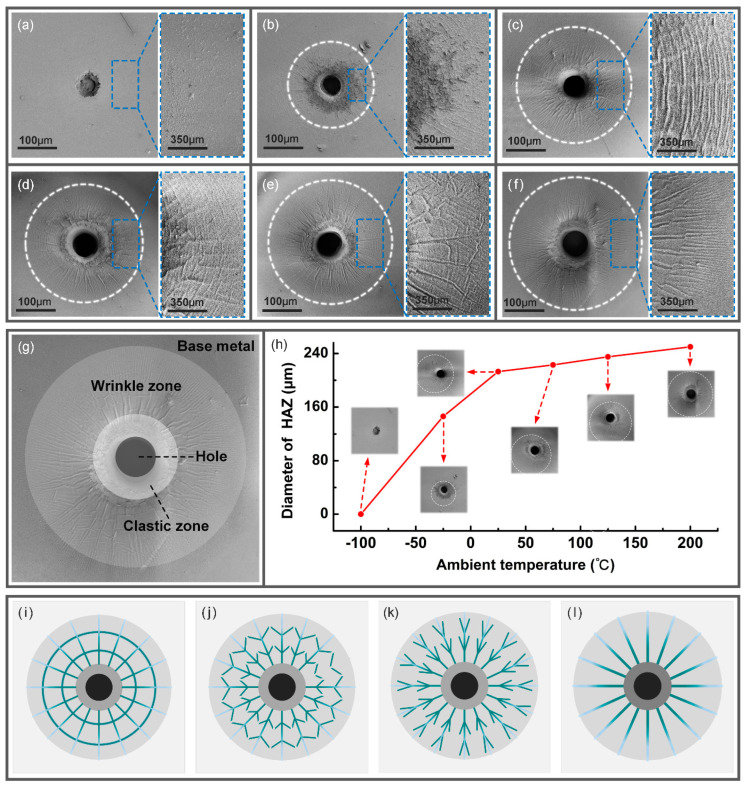
The evolution of HAZ with ambient temperature. (**a**–**f**) SEM images of HAZ morphology when the ambient temperature is −100 °C, −25 °C, 25 °C, 75 °C, 125 °C and 200 °C, respectively. (**g**) The composition of HAZ. (**h**) The evolution of the HAZ diameter with ambient temperature. (**i**–**l**) Schematic diagram of the evolution of wrinkles with ambient temperature.

**Figure 9 micromachines-14-00090-f009:**
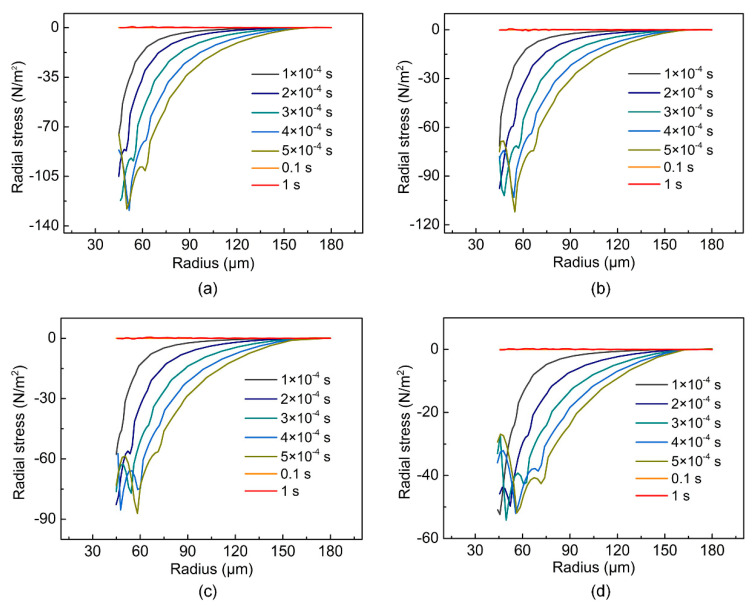
Stress distribution along the radius direction of upper surface at different moments when the ambient temperature is (**a**–**d**) 25 °C, 75 °C, 125 °C and 200 °C, respectively.

**Figure 10 micromachines-14-00090-f010:**
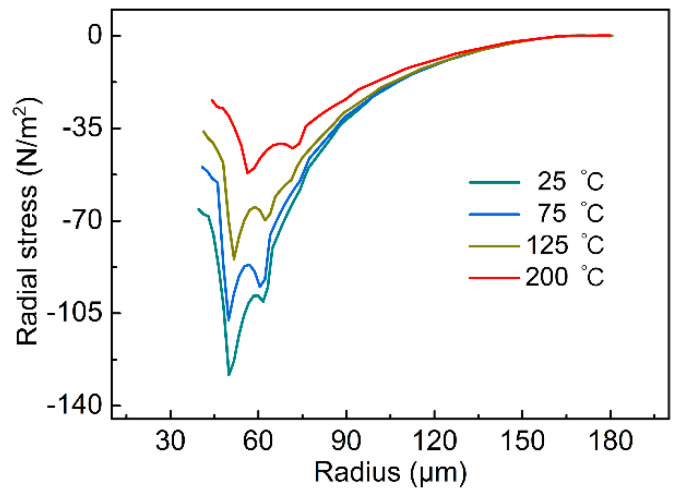
Stress distribution along the radius direction of upper surface at different ambient temperatures.

**Table 1 micromachines-14-00090-t001:** Laser micro-drilling parameters and their values.

Parameters	Values
Ambient temperature (°C)	−100, −125, −25, 25, 75, 125, 200
Laser fluence (J/cm²)	18.87
Laser frequency (KHz)	10
Pulse duration (s)	5 × 10^−4^

**Table 2 micromachines-14-00090-t002:** Simulation parameters and their values.

Parameters	Symbol	Values
Output laser fluence (W)	*Q* _0_	0.8
Laser spot radius (μm)	*r*	40
Pulse duration (s)	*t_s_*	5 × 10^−4^
Laser frequency (kHZ)	*f*	10
Density (kg/m^2^)	ρ	980
Specific heat J/(kg·K)	*C_P_*	1465
Ambient temperature (°C)	*T_set_*	−100, −125, −25, 25, 75, 125, 200
Thermal conductivity [W/(m·k)]	*k*	0.17
Surface emissivity	*ε*	0.8
Poisson’s ratio	*ν*	0.49
Young’s modulus (Mpa)	*E*	2.3
Linear expansion coefficient (W/k)	*α*	9 × 10^−4^

## Data Availability

No new data were created in this study.

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
