# Peer review of "Effects of Ambient Temperature on Nanosecond Laser Micro-Drilling of Polydimethylsiloxane (PDMS)"

_micromachines, 2022, doi:10.3390/mi14010090_

Round 1

Reviewer 1 Report

In this manuscript, the influences of temperature on nanosecond laser micro-drilling of PDMS were investigated by simulation and experiment. The results of simulation and experiment are almost identical, which provides a good guide for the experiment. The method proposed in this work could find some applications for engineering application of PDMS. However, some revisions are needed to improve the manuscript. Following are several questions for the authors:

1.        For the investigation of the cross-sectional morphology fabricated by laser, normally the milling machine was used to polish the cross section to observe the depth and taper of the hole. The cross-section of the micro-hole should be presented by SEM to obtain the geometrical characteristics such as the simulation results in Figure 4.

2.        The geometric characteristics and the heat affected zone of PDMS micro-hole influenced by temperature were studied. Why the authors chose temperature for this research? It is suggested that the authors explain in detail the purpose of the study and clearly state what are the advancements in this work.

3.        In Figure 7, how many groups of data did the authors calculate when conducting the statistics of the relationship between diameter and taper with temperature? To avoid the random error of the experimental data, multiple sets of data should be measured; the average and data error should be calculated, and represented as error bars in the graph.

4.        Is there any oxide or other impurities generated around the micro-hole induced by nanosecond laser?

5.        There are many literatures on the on the ultrafast laser-material interactions. The author should compare its advantages and disadvantages with the literature, such as:

[1] Y. Song, C. Wang, X. Dong, K. Yin, F. Zhang, Z. Xie, D. Chu, J. Duan, Controllable superhydrophobic aluminum surfaces with tunable adhesion fabricated by femtosecond laser, Opt. Laser Technol. 102 (2018) 25-31.

[2] Jia X.S., Chen Y.Q., er al, Combined pulse laser: Reliable tool for high-quality, high-efficiency material processing, Opt. Laser Technol., 2022, 153:108209.

[3] K. Ding, C. Wang, S. Li, X. Zhang, N. Lin, J. Duan, Large-area cactus-like micro-/nanostructures with anti-reflection and superhydrophobicity fabricated by femtosecond laser and thermal treatment, Surf. Interfaces 33 (2022) 102292.

[4] Ding K.W., Wang C., Li S.H., Zhang X.F., Lin N. and Duan J.A., Single-step femtosecond laser structuring of multifunctional colorful metal surface and its origin, Surf. Interfaces, 2022, 34:102386.

Author Response

Dear reviewer,

Thanks for your comments to our manuscript entitled “Effects of ambient temperature on nanosecond laser mi-cro-drilling of polydimethylsiloxane (PDMS)”. We are very grateful for your and all reviewers’ efforts and constructive comments. We have carefully considered all the comments of the reviewers and revised our manuscript accordingly.

Below we provide our point-by-point reply and indicate where we revised our manuscript. We believe that with a more detailed analysis, the manuscript has been strengthened.

Finally, we appreciate very much for your time in editing our manuscript and your valuable suggestions. I am looking forward to hearing from your decision when it is made.

Prof. Dr. Congyi Wu

Department of Mechanical science and Engineering

Huazhong University of Science and Technology

E-mail: [email protected]  Tel: +86-027-87559835

#Reviewer 1

In this manuscript, the influences of temperature on nanosecond laser micro-drilling of PDMS were investigated by simulation and experiment. The results of simulation and experiment are almost identical, which provides a good guide for the experiment. The method proposed in this work could find some applications for engineering application of PDMS. However, some revisions are needed to improve the manuscript. Following are several questions for the authors: 

  1. For the investigation of the cross-sectional morphology fabricated by laser, normally the milling machine was used to polish the cross section to observe the depth and taper of the hole. The cross-section of the micro-hole should be presented by SEM to obtain the geometrical characteristics such as the simulation results in Figure 4.

Author reply:

Thanks for your suggestion and your consideration is very comprehensive. Every comment and suggestion you provide is of great significance to the further modification of our manuscript and subsequent research work. Please allow me to respond to relevant questions in detail.

As PDMS is a soft material, its cross-sectional morphology is difficult to observe, so this study takes the surface morphology change as the entry point to analyze the effect of laser process parameters on micro-hole morphology.

  1. The geometric characteristics and the heat affected zone of PDMS micro-hole influenced by temperature were studied. Why the authors chose temperature for this research? It is suggested that the authors explain in detail the purpose of the study and clearly state what are the advancements in this work.

Author reply:

Thank you for your professional questions.

Through a large number of preliminary experiments, it was found that the process of PDMS removal is dominated by thermal ablation. In this process, the laser generates heat, the PDMS absorbs heat and etches, and the heat diffuses in the body to form a heat-affected zone. Therefore, it’s believed that the HAZ can be reduced by reducing the ambient temperature to slow down heat diffusion. The experimental results also show that the HAZ can be significantly reduced by lowering the ambient temperature. As shown in Figure 5, the HAZ basically disappears under the same process parameters after the ambient temperature is reduced to -100°C.

Based on your suggestions, we have made further additions to the purpose of the study and the progress of the work.

The detailed modifications are listed as below:

(Page 2, Line 68) “In this paper, the effects of ambient temperature (-100℃~200℃) on the geometric characteristics and the heat affected zone (HAZ) of PDMS micro-hole processed by nanosecond ultraviolet (UV,355nm) laser were studied.” Modified to “In this paper, the effects of ambient temperature (-100℃~200℃) on the geometric characteristics and the heat affected zone (HAZ) of PDMS micro-hole processed by nanosecond ultraviolet (UV,355nm) laser were studied. The experimental results show that under the same process parameters, lowering the ambient temperature can effectively reduce the size of HAZ and thus improve the process quality.”

  1. In Figure 7, how many groups of data did the authors calculate when conducting the statistics of the relationship between diameter and taper with temperature? To avoid the random error of the experimental data, multiple sets of data should be measured; the average and data error should be calculated, and represented as error bars in the graph.

Author reply:

Thank you very much for your professional and considerate advice.

As described in the main text, to ensure the reproducibility of the experiments, each set of experiments was repeated 10 times at the same ambient temperature, and 10 PDMS samples were obtained for each set. Further, we compared the average of the experimental data with the simulated values. Although the experimental data have error bars, the introduction of experimental data error bars introduces greater presentation ambiguity when the average of the experimental data is compared with the simulated values. Therefore, after further consideration, we believe that the relationship between the experimental data and the simulated data can be better expressed without adding error bars here.

However, based on your feedback, we have provided further clarification of the data in the manuscript.

The detailed modifications are listed as below:

(Page 10, Line 289) “The maximum prediction errors (Emax) of top micro-hole diameter, bottom mi-cro-hole diameter and micro-hole taper were 2.785%, 6.306% and 9.688%, which indicated that the simulation results showed a good agreement with the experiment results in in mi-cro-hole geometry.” Modified to “The maximum prediction errors (Emax) of top micro-hole diameter, bottom mi-cro-hole diameter and micro-hole taper were 2.785%, 6.306% and 9.688%, compared with the average value of 10 sets of data for each sample, which indicated that the simulation results showed a good agreement with the experiment results in in mi-cro-hole geometry.”

  1. Is there any oxide or other impurities generated around the micro-hole induced by nanosecond laser?

Author reply:

Your query is very professional.

In order to investigate the interaction process between laser and PDMS, we have done a lot of research on the reaction products in the early stage. The results showed that the solid products around the micropores are all silica.

[1] Y. Rong, Y. Huang, C. Lin, Y. Liu, S. Shi, G. Zhang, C. Wu*. Stretchability improvement of flexiable electronics by laser micro-drilling array holes in PDMS film. Optics and Lasers in Engineering, 134 (2020).

[2] C. Wu, J. Xu, T. Zhang, G. Xin, M. Li, Y. Rong, G. Zhang, Y. Huang. Precision cutting of PDMS film with UV-nanosecond laser based on heat generation-diffusion regulation. Optics and Laser Technology, 145 (2022).

  1. There are many literatures on the on the ultrafast laser-material interactions. The author should compare its advantages and disadvantages with the literature, such as:

[1] Y. Song, C. Wang, X. Dong, K. Yin, F. Zhang, Z. Xie, D. Chu, J. Duan, Controllable superhydrophobic aluminum surfaces with tunable adhesion fabricated by femtosecond laser, Opt. Laser Technol. 102 (2018) 25-31.

[2] Jia X.S., Chen Y.Q., er al, Combined pulse laser: Reliable tool for high-quality, high-efficiency material processing, Opt. Laser Technol., 2022, 153:108209.

[3] K. Ding, C. Wang, S. Li, X. Zhang, N. Lin, J. Duan, Large-area cactus-like micro-/nanostructures with anti-reflection and superhydrophobicity fabricated by femtosecond laser and thermal treatment, Surf. Interfaces 33 (2022) 102292.

[4] Ding K.W., Wang C., Li S.H., Zhang X.F., Lin N. and Duan J.A., Single-step femtosecond laser structuring of multifunctional colorful metal surface and its origin, Surf. Interfaces, 2022, 34:102386.

Author reply:

Thanks for your suggestion and your consideration is very comprehensive.

Based on your suggestion, we have introduced the suggested literature and added a comparative introduction in the introduction section.

The detailed modifications are listed as below:

(Page 1, Line 34) “Compared to these processing methods, laser processing is a more suitable method, which has been widely used in aerospace [10], automotive industry [11], elec-tronic industry [12]and other fields [13].” modified to “Compared with these processing methods, laser processing is a more efficient and convenient processing method[10], and it has been widely used for the preparation of superhydrophobic microstructures[11], bionic microstructures[12], phantom microstructures[13], etc.”

Reviewer 2 Report

The authors described the effect of ambient temperature on UV laser drilling process for PDMS material. After carefully reading the manuscript, however, I have several concerns regarding to the manuscript listed as follows: 

  1. The goal of this research was not clearly understood described in “Introduction” section. For example, is changing the ambient temperature practical? In what circumstance would the changing of the ambient temperature in laser drilling process be interested?
  2. In Line 71: it seems that part of the sentence, “changed with change with the change of….,” has some grammar errors.  
  3. In the manuscript, “HZ” should be changed to “Hz.”
  4. In the manuscript, “formula” is suggested to change to “equation.”
  5. In Line 170: “α” was not found in equation (5).
  6. In Figure 7, Emax should be expressed in percentage, i.e. add “%” in the figure.
  7. From Line 339 to Line345: “t=5e-4s” should be changed to “t = 5 x 10-4 s”
  8. In Figure 9 and Figure10, were the stress distribution curved obtained by experiments? If so, the experimental setup should be described in the manuscript. Otherwise, the calculation or simulation process should be described in the manuscript instead.

Author Response

Dear reviewer,

Thanks for your comments to our manuscript entitled “Effects of ambient temperature on nanosecond laser mi-cro-drilling of polydimethylsiloxane (PDMS)”. We are very grateful for your and all reviewers’ efforts and constructive comments. We have carefully considered all the comments of the reviewers and revised our manuscript accordingly.

Below we provide our point-by-point reply and indicate where we revised our manuscript. We believe that with a more detailed analysis, the manuscript has been strengthened.

Finally, we appreciate very much for your time in editing our manuscript and your valuable suggestions. I am looking forward to hearing from your decision when it is made.

Prof. Dr. Congyi Wu

Department of Mechanical science and Engineering

Huazhong University of Science and Technology

E-mail: [email protected]  Tel: +86-027-87559835

#Reviewer 2

The authors described the effect of ambient temperature on UV laser drilling process for PDMS material. After carefully reading the manuscript, however, I have several concerns regarding to the manuscript listed as follows: 

  1. The goal of this research was not clearly understood described in “Introduction” section. For example, is changing the ambient temperature practical? In what circumstance would the changing of the ambient temperature in laser drilling process be interested?

Author reply:

Thanks for your suggestion and your consideration is very comprehensive. Every comment and suggestion you provide is of great significance to the further modification of our manuscript and subsequent research work. Please allow me to respond to relevant questions in detail.

Lowering the ambient temperature effectively reduces the size of HAZ, which in turn improves process quality. Through a large number of preliminary experiments, it was found that the process of PDMS removal is dominated by thermal ablation. In this process, the laser generates heat, the PDMS absorbs heat and etches, and the heat diffuses in the body to form a heat-affected zone. Therefore, it’s believed that the HAZ can be reduced by reducing the ambient temperature to slow down heat diffusion. The experimental results also show that the HAZ can be significantly reduced by lowering the ambient temperature. As shown in Figure 5, the HAZ basically disappears under the same process parameters after the ambient temperature is reduced to -100°C.

[1] Y. Rong, Y. Huang, C. Lin, Y. Liu, S. Shi, G. Zhang, C. Wu*. Stretchability improvement of flexiable electronics by laser micro-drilling array holes in PDMS film. Optics and Lasers in Engineering, 134 (2020).

[2] C. Wu, J. Xu, T. Zhang, G. Xin, M. Li, Y. Rong, G. Zhang, Y. Huang. Precision cutting of PDMS film with UV-nanosecond laser based on heat generation-diffusion regulation. Optics and Laser Technology, 145 (2022).

  1. In Line 71: it seems that part of the sentence, “changed with change with the change of….,” has some grammar errors.  

Author reply:

We apologize for the trouble you had reading this due to an inaccurate description. Based on your suggestion, we have revised the description.

The detailed modifications are listed as below:

(Page 2, Line 75) “Aiming at the phenomenon that the HAZ observed in the experiment changed with change with the change of the ambient temperature, the evolution of the wrinkles in the HAZ is explained by solving the transient stress distribution.” modified to “Aiming at the phenomenon that the HAZ observed in the experiment changed with the change of the ambient temperature, the evolution of the wrinkles in the HAZ is explained by solving the transient stress distribution.”

  1. In the manuscript, “HZ” should be changed to “Hz.”

Author reply:

Based on your suggestion, we have revised the description.

The detailed modifications are listed as below:

(Page 3, Line 108),(Page 3, Table 1),(Page 6, Table 2) “kHZ” modified to “kHz”

  1. In the manuscript, “formula” is suggested to change to “equation.”

Author reply:

Based on your suggestion, we have revised the description.

The detailed modifications are listed as below:

(Page 4, Line 129),(Page 4, Line 135),(Page 5, Table 186) “formula” modified to “equation”.

  1. In Line 170: “α” was not found in equation (5).

Author reply:

α represents the linear expansion coefficient, which begins to appear in Equation 9 and Equation 10 when calculating the stress-strain relationship.

  1. In Figure 7, Emax should be expressed in percentage, i.e. add “%” in the figure.

Author reply:

Thank you for the correction.

Based on your suggestions, we have corrected Figure 7

The detailed modifications are listed as below:

(Page 10, Figure 7) modified to

Figure 7. (a) Experimental and simulation diameter of the top and bottom micro-hole at different ambient temperatures. (b) Experimental and simulation taper of the micro-hole at different ambient temperatures.

  1. From Line 339 to Line345: “t=5e-4s” should be changed to “t = 5 x 10-4 s”

Author reply:

Thank you for your professional advice. Based on your suggestion, we have revised the description.

The detailed modifications are listed as below:

(Page 12, Line 348),(Page 12, Line 349),(Page 13, Table 351) “t=5e-4s” modified to “t = 5 x 10-4 s”.

  1. In Figure 9 and Figure10, were the stress distribution curved obtained by experiments? If so, the experimental setup should be described in the manuscript. Otherwise, the calculation or simulation process should be described in the manuscript instead.

Author reply:

I'm sorry for the difficulty in reading because of unclear expressions.

The stress distribution curves in Figures 9 and 10 are obtained by simulation, and the simulation-related parameters are described in detail in the Thermo-mechanical modeling section of the manuscript.

Based on your feedback, we have illustrated the results of Figure 9 and Figure 10.

The detailed modifications are listed as below:

(Page 12, Line 345) “This evolution trend was related to the maximum thermal stress in the radial direction of PDMS at different ambient temperatures.” modified to “The model results show that this evolutionary trend is related to the maximum thermal stress in the PDMS radial direction at different ambient temperatures.”

Round 2

Reviewer 2 Report

The authors have responded to all my concerns regarding to this manuscript adequately. There are still two minor corrections should be made before the acceptance for publication.

1.     In Line 188, the description of “α” should be moved to after Equation (10).

2.     From Line 363 to Line 367: the power of 10 should be in superscript.